# Dissolution Behavior of Lime with Different Properties into Converter Slag

**DOI:** 10.3390/ma16196487

**Published:** 2023-09-29

**Authors:** Mengxu Zhang, Jianli Li, Cao Jia, Yue Yu

**Affiliations:** 1The State Key Laboratory of Refractories and Metallurgy, Wuhan University of Science and Technology, Wuhan 430081, China; zmengxu@wust.edu.cn (M.Z.); jiacao@wust.edu.cn (C.J.); yuyue@wust.edu.cn (Y.Y.); 2Hubei Provincial Key Laboratory for New Processes of Ironmaking and Steelmaking, Wuhan University of Science and Technology, Wuhan 430081, China

**Keywords:** properties of quicklime, reaction interface, 2CaO·SiO_2_, dissolution rate, mass transfer

## Abstract

China’s 2022 crude steel production soared to an impressive 1.018 billion tons, and steel slag constituted approximately 10% to 15% of this massive output. However, a notable hindrance to the comprehensive utilization of steel slag arises from the fact that it contains 10% to 20% of free calcium oxide (f-CaO), resulting in volume instability. To address this challenge, our study delved into the dynamic transformation of the interface between lime and slag, as well as the fluctuations in the dissolution rate of lime. An Electron Probe Micro Analyzer, equipped with an energy-dispersive spectrometer, was employed for the analysis. Our findings revealed that the configuration of the reaction interface between quicklime and slag underwent alterations throughout various phases of converter smelting. At a temperature of 1400 °C, several significant transformations occurred, including the formation of a CaO-FeO solid solution, (Ca, Mg, Fe) olivine, and low-melting point (Ca, Mg) silicate minerals. With the gradual reduction in FeO content, a robust and high-melting 2CaO·SiO_2_ layer emerged, generated through the interaction between CaO and (Ca, Mg, Fe) olivine. Furthermore, for lime with a particle size of 20 mm and a calcination rate of 0%, the thickest layer of 2CaO·SiO_2_ was observed after 120 s of dissolution in slag A2 at 1400 °C. Overall, the dissolution rates of lime with different particle sizes in slag A1 to A4 showed a gradual increase. On the other hand, the dissolution rates of lime with different calcination rates in slag A1 to A4 exhibited an initial increase, followed by a decrease, and then another increase. The formation of a high-melting point and continuous dense 2CaO·SiO_2_ layer during the dissolution process hindered the mass transfer between lime and slag.

## 1. Introduction

In 2022, China achieved a momentous feat in its crude steel production, reaching an astounding 1.018 billion tons. However, given its status as a substantial consumer and emitter of fossil fuels, the iron and steel industry has become a focal point for endeavors aimed at reducing carbon emissions [1]. Considering China’s dominance in the global crude steel production, with a share exceeding 50%, the carbon emissions emanating from China’s iron and steel industry far exceed 60% of the global total and approximately 15% of the national total. Consequently, this sector stands as the largest emitter among the 31 manufacturing categories, necessitating the implementation of strategies aimed at enhancing the comprehensive utilization of steelmaking slag, fostering the development of environmentally friendly and low-carbon steelmaking technologies, and establishing novel approaches for low-carbon steel production. These endeavors play a critical role in effectively reducing the carbon emissions associated with the steel industry [2]. Lime assumes a pivotal role as one of the fundamental materials in the slag formation process during steelmaking. Typically, lime particles exhibit a particle size ranging from 10 to 50 mm, with an addition rate of 20–40 kg per metric ton of molten iron. Steel slag, a byproduct arising from the steelmaking process, presents notable environmental hazards when amassed in substantial quantities. Typically, steel slag contains a mass fraction of calcium oxide (CaO) within the range of 40% to 50%, frequently showcasing discernible white or brownish spots commonly known as f-CaO. This fraction of f-CaO can induce volume instability and impede the comprehensive utilization of steel slag [3].

The advancement of expeditious lime dissolution and its integration into slag represents a highly significant and extensively researched domain. Matsushima et al. [4] conducted an in-depth examination of lime particle dissolution in slag using the rotating cylinder method. Their findings revealed that as the rotation speed increased, the rate of lime dissolution also escalated, displaying a linear correlation with time. Additionally, they submerged lime within static CaO-FeO-Al_2_O_3_-based slag to investigate the distribution of slag components proximate to the interface and the formation of 2CaO·SiO_2_, referred to as C2S hereafter. Interestingly, the precipitation of C2S was observed at a distance from the interface. In parallel research, D. Sichen et al. [5] delved into the static dissolution of lime within two distinct systems: FeO-SiO_2_ and CaO-FeO-SiO_2_ slags. Their work emphasized that lime exhibited a heightened solubility within the FeO-SiO_2_ system, while in the CaO-FeO-SiO_2_ system, a dense layer of C2S emerged. This phenomenon was attributed to the increased dissolution driving force of CaO within the FeO-SiO_2_ system, and the formation of the C2S layer was exclusively observed under conditions of low FeO content. Furthermore, Xia et al. [6] explored the dissolution behavior of lime particles within slag systems enriched with P_2_O_5_. Their investigation revealed that in the CaO-FeO-SiO_2_-P_2_O_5_ system, the interface exhibited the formation of a solid solution composed of 3CaO·P_2_O_5_ instead of the customary C2S layer. However, it is noteworthy that the creation of this solid solution also impeded the dissolution of lime. Lastly, Ludimila et al. [7] embarked on a study focused on the impact of lime particle size on its dissolution within slag. Their observations demonstrated that an increase in lime particle size resulted in a reduction in dissolution rate, primarily due to the diminished contact area between lime and slag.

The development of a compact C2S layer hinders the dissolution of lime by deteriorating the kinetic conditions, ultimately leading to a surge in undissolved lime particles during the latter phases of converter slag processing. This limitation presents a challenge for achieving high recycling rates of undissolved lime particles in converter slag. The dissolution of lime is influenced by both the physical structure of lime itself and the properties of the slag [3,4,5,6,7]. During the formation of calcium-bearing slag in the converter, the interface between the initial slag and quicklime serves as an indicator of the lime dissolution progress. However, there is limited research regarding the impact of quicklime dissolution time in slag on its dissolution rate. To attain a comprehensive grasp of the lime dissolution process, this study seeks to investigate the interface reactions and dissolution rates of lime with distinct characteristics in slags of varying compositions at the onset of converter smelting. This research aims to shed light on the evolutionary changes in the reaction interface and the dissolution mechanism of quicklime within converter slag. Such insights will establish a theoretical foundation for the swift formation of slag in the converter.

## 2. Experimental Procedure

### 2.1. Raw Materials

The raw material used in this experiment is laboratory-calcined quicklime, which is derived from limestone provided by a large domestic steel company. The composition of the limestone was determined by using X-ray fluorescence (XRF) analysis, which showed that the main component of the limestone is CaCO_3_, with a content of 98.55%. It also contains small amounts of MgO, SiO_2_, and other components. X-ray diffraction analysis using a PHILIPS-Xpertpro instrument confirmed that calcite (CaCO_3_) is the only mineral phase present in the limestone. The slag system studied in this experiment corresponds to the early to middle stage of the slag-forming route based on the CaO component in converter smelting, with the slag composition ratios shown in Table 1. CaO, SiO_2_, and MgO were used as chemically pure reagents, while FeO was added in the form of ferrous oxalate. The corresponding points of the synthetic slag composition in the ternary phase diagram are shown in Figure 1.

### 2.2. Experimental Procedure

The experimental procedure encompasses several sequential stages, including the preparation of cylindrical lime, synthesis of the slag, dissolution experiments, and sample analysis. The initial step involves the preparation of cylindrical limestone particles, and the detailed process is as follows: First, raw limestone was initially sliced into small pieces using a cutting machine. Subsequently, the limestone blocks were ground into cylindrical shapes, each with heights of 15 mm and diameters ranging from 10 mm, 15 mm, 20 mm, to 25 mm, respectively, utilizing a grinding machine. These cylindrical limestone specimens were then meticulously cleansed and subjected to drying in an oven at a temperature of 200 °C for a duration of three hours. After the drying process, the limestone samples were allowed to cool to room temperature and were preserved in a vacuum container for future use. The next step involved the utilization of a carbon tube furnace, with high-purity N2 gas serving as a protective atmosphere. This furnace was gradually heated at a specified heating rate until it reached a temperature of 1350 °C. The previously prepared cylindrical lime particles were placed within a molybdenum basket, strategically located within the constant temperature zone of the carbon tube furnace. Initially, all lime particles, regardless of their size, were subjected to a 10 min heating period to facilitate the initial formation. Subsequently, the 15 mm lime particles underwent distinct heating durations to achieve varying calcination degrees, specifically 0%, 44%, 60%, and 84%, respectively. Finally, the calcined lime was cooled to room temperature and stored in a vacuum desiccator for subsequent use. The calcination rate is defined according to Equation (1):(1)$$\varphi =\left(1-\frac{m_{1}-m_{2}}{0.44\;m_{1}}\right)\times 100\%$$
where φ is the calcination rate of lime, $m_{1}$ is the mass of the limestone, expressed in g, and $m_{2}$ is the mass of calcined lime, expressed in g.

In the second phase of the experiment, the synthetic slag was meticulously prepared in accordance with the composition outlined in Table 1. High-purity reagents were employed and subjected to multiple rounds of sieving to ensure thorough mixing and homogeneity. The thoroughly blended slag mixture was subsequently introduced into a corundum crucible and compacted to promote an effective calcination of the powdered reagents. The corundum crucible, now loaded with the slag, was positioned within the constant temperature zone of the carbon tube furnace. The temperature was incrementally raised to 1100 °C at a rate consistent with the furnace’s heating capacity. Once this temperature was reached, it was maintained for a duration of 30 min to facilitate the desired reactions. Subsequently, the crucible was carefully removed from the furnace and allowed to cool to room temperature, rendering it ready for subsequent utilization in the experiment. The third step of the experiment focused on conducting the dissolution experiments, and the experimental setup is depicted in Figure 2. To initiate the experiments, the pre-melted slag was first crushed into smaller particles and then carefully loaded into a magnesia crucible. Following the preparation of the magnesia crucible and its loading with pre-melted slag, the crucible was positioned within the constant temperature zone of the carbon tube furnace. A high-purity N_2_ gas, flowing at a rate of 10 L/min, was introduced into the furnace to maintain a protective atmosphere. The temperature within the furnace was then gradually increased at a rate of 10 °C/min until it reached the target temperature of 1400 °C. Upon reaching this temperature, the system was allowed to equilibrate for 5 min, ensuring the complete melting of the slag. Subsequently, the prepared lime was introduced into the molten slag. After a dissolution period of 120 s, the lime particles were carefully extracted from the slag and rapidly cooled to room temperature through air cooling for further analysis.

The concluding step of the experiment entails the analysis of the lime samples. The cooled lime samples were cut in half, partially embedded using a mounting press, and then subjected to grinding, polishing, and cleaning with an ultrasonic cleaning device. Subsequently, the samples were dried in a vacuum desiccator and underwent gold sputtering for scanning electron microscopy observation of the microstructure at the lime–slag reaction interface. The collected samples were analyzed using a field emission electron probe microanalyzer (EPMA 8050G) to observe the microstructure and a configured energy dispersive spectrometer (AZtecLive UltimMax 40) to determine the composition of the microregions. The average thickness of each phase layer in the slag was measured using ImageJ software.

## 3. Experimental Results

### 3.1. Variation in Lime Particle Diameter after Dissolution in Slag for Different Particle Sizes and Calcination Rates

The variation in lime particle diameter after dissolution in slag for different particle sizes and calcination rates is presented in Table 2 and Table 3, respectively. The tables clearly illustrate that lime particles with varying sizes and calcination rates underwent a reduction in radius following a 120 s dissolution period in the different slags A1 to A4. For this experiment, the lime particles were assumed to be cylindrical in shape, and their volume (V) and surface area (S) can be calculated using Equations (2) and (3):(2)$${V=\pi r}^{2}h$$
(3)S=2πrh
where r is the radius of the cylindrical lime particle, expressed in m, and h is the height of cylindrical lime particles, expressed in m.

The dissolution rate of quicklime in the slag, denoted as $V_{r}$, can be described as the volume of dissolution per unit area within a specified time interval [8]. This rate is expressed as in Equation (4):(4)$$V_{r}=-\frac{1}{S}\frac{\mathrm{dV}}{\mathrm{dt}}=-\frac{\mathrm{dr}}{\mathrm{dt}}$$
where $V_{r}$ is the dissolution rate of the lime, expressed in m/s, dr is the decrease in lime radius, expressed in m, and dt is the dissolution time, expressed in s.

By substituting the data from Table 2 and Table 3 into the aforementioned equations, the relationship between the dissolution rate of lime with different properties and the composition of slag A1 to A4 can be obtained. This relationship is depicted in Figure 3 and Figure 4. From Figure 3, it can be observed that the dissolution rate of lime particles with different sizes increases with the increase in particle size in various slags after a dissolution time of 120 s. However, the growth trend in slag A2 was different from that in the other slags. In slag A2, the dissolution rate of lime particles initially increased slowly with the increase in particle size, then rapidly increased, and finally reached a maximum value of 3.87 × 10^−6^ m/s. In slags A1, A3, and A4, the dissolution rate of lime particles showed a trend of rapid increase, followed by a slow increase, and then another rapid increase with the increase in particle size. Specifically, in slag A1, the dissolution rate of lime increased gradually from 1.44 × 10^−6^ m/s to 3.33 × 10^−6^ m/s as the particle size increased. In slag A3, the dissolution rate increased from 2.51 × 10^−6^ m/s to 4.37 × 10^−6^ m/s, and in slag A4, it increased from 1.39 × 10^−6^ m/s to 3.51 × 10^−6^ m/s. When the lime particle size was 10 mm, the maximum dissolution rate was observed in slag A2, with a value of 2.51 × 10^−6^ m/s. When the lime particle size was 15 mm, the dissolution rates in different slags were relatively close, with the maximum rate of 2.51 × 10^−6^ m/s in slag A3 and the minimum rate of 1.44 × 10^−6^ m/s in slag A1. The largest difference in dissolution rates occurred when the lime particle size was 20 mm. In this case, the dissolution rates in the slags followed the order of A2 > A3 > A4 > A1, and the difference between the maximum and minimum rates was 0.8 × 10^−6^ m/s. When the lime particle size was 25 mm, the dissolution rates in the slags were in the order of A3 > A2 > A4 > A1, with a maximum rate of 4.37 × 10^−6^ m/s. From Figure 4, it can be observed that the dissolution rates of lime with different calcination rates in various slags exhibit a similar trend of initially increasing, then decreasing, and finally increasing again after a dissolution time of 120 s. Although the trend of dissolution rate variation was the same, the calcination rate at which the dissolution rate reached its maximum value differed slightly. When the calcination rate was 60%, the dissolution rate of lime in different slags was at its minimum. However, the calcination rate at which the lime dissolution rate reached its maximum value varies in different slags. In slags A1, A3 and A4, the lime dissolution rate reached its maximum value at a calcination rate of 84%, with values of 2.38 × 10^−6^ m/s, 2.59 × 10^−6^ m/s, and 4.30 × 10^−6^ m/s, respectively. However, in slag A2, the lime dissolution rate reached its maximum value of 2.91 × 10^−6^ m/s at a calcination rate of 44%. Additionally, the growth in lime dissolution rate was most significant in the range of 60% to 84% calcination rate. Among them, when lime with an 84% calcination rate was dissolved in slag A3, it exhibited the highest growth in dissolution rate, increasing from 1.00 × 10^−6^ m/s to 4.30 × 10^−6^ m/s. The order of lime dissolution rates in this case was A3 > A2 > A4 > A1, which followed the same pattern observed for lime particles with a particle size of 25 mm.

### 3.2. Evolution of the Reaction Interface between Lime and Slag during Static Dissolution with Different Particle Sizes

Figure 5 presents the microstructure of the lime–slag reaction interface after 120 s of reaction at 1400 °C for different particle sizes of lime and different slags. It can be observed that a layered structure was formed at the reaction interface during the lime–slag reaction. The main phases included the (Ca, Mg, Fe) olivine layer, C2S layer, and CaO-FeO solid solution layer. The mean thickness of the solid solution layer that developed at the reaction interface with different lime particle sizes was computed and subjected to statistical analysis using ImageJ, as presented in Figure 6. As depicted in Figure 6, it is evident that in the case of slags A1 and A3, the thickness of the solid solution layer exhibited a gradual reduction followed by an increase with the augmentation of lime particle size. However, in slags A2 and A4, the average thickness of the solid solution layer exhibited a decreasing–increasing–decreasing trend. In slag A1, the maximum thickness of the solid solution layer was 13.17 μm when the lime particle size was 25 mm. In slags A2 to A4, the maximum thickness of the solid solution layer was obtained when the lime particle size was 10 mm, measuring 15.72 μm, 15.14 μm, and 15.94 μm, respectively. Additionally, it is worth noting that the mean thickness of the solid solution layer reached its lowest point at 3.98 μm when reacting with lime particles measuring 15 mm in slag A3. When the lime particle size was 15 mm, the average thickness of the solid solution layer formed during the reaction with slag followed the sequence A2 > A1 > A4 > A3. When the lime particle size was 20 mm, the order was A2 > A4 > A1 > A3.

The evolution of the average thickness of the (Ca, Mg, Fe) olivine layer formed at the lime–slag reaction interface during the 1400 °C reaction, involving lime particles of different sizes and various slags, is graphically represented in Figure 7. The observations from Figure 7 clearly demonstrate that, across the four distinct slag compositions, the average thickness of the (Ca, Mg, Fe) olivine layer initially increased and then decreased as the size of the lime particles increased. Remarkably, the trends in the variation in the (Ca, Mg, Fe) olivine layer thickness in slags A1 to A4 were consistent, with all reaching their maximum values when the lime particle size was 20 mm before subsequently decreasing. The respective maximum values were 122.402 μm, 118.142 μm, 108.032 μm, and 98.182 μm.

The microstructure of the reaction interface when lime particles with a diameter of 10 mm reacted with different slags at 1400 °C is shown in Figure 8. It can be observed from the SEM-EDS analysis that C2S was formed, but no C2S layer was formed in any of the slags. The changes in the average thickness of the C2S layer at the lime–slag reaction interface, in response to an increase in lime particle size, are visually presented in Figure 9. It was evident from Figure 9 that when the lime particle size was 15 mm, only slag A3 did not form a C2S layer, which may be related to the viscosity of slag A3. Furthermore, the thickness of the C2S layer formed during the lime–slag reaction in slags A1 to A4 increased slowly and then decreased slowly with an increasing lime particle size. The maximum thickness of the C2S layer was observed when the lime particle size was 20 mm, with values of 6.49 μm, 12.11 μm, 4.17 μm, and 4.81 μm for slags A1 to A4, respectively. Overall, the thickness of the C2S layer followed the trend A2 > A1 > A4 > A3.

### 3.3. Evolution of the Reaction Interface between Lime and Slag during Static Dissolution with Different Calcination Rates

Figure 10 displayed the microstructural variations at the reaction interface between different limestone calcination rates and various slags during the static dissolution at 1400 °C for 120 s. It was evident that a stratified structure formed at the reaction interface. The average thickness of the solid solution layer formed at the reaction interface, as influenced by the calcination rate of the limestone, is depicted in Figure 11. From the data presented in Figure 11, it is evident that in slag A1, the thickness of the solid solution layer exhibited a gradual increase with an escalation in the calcination rate. It reached its maximum value of 23.48 μm before subsequently declining. In contrast, for slag A2, the thickness of the solid solution layer displayed an initial decrease, followed by a gradual increase, with the maximum value of 24.25 μm being attained at a calcination rate of 84%. In slags A3 and A4, the average thickness of the solid solution layer first increased, then decreased, and finally increased again. The maximum values were observed at a limestone calcination rate of 44%, measuring 10.84 μm and 14.86 μm, respectively. Therefore, the order of maximum thickness of the solid solution layer was A4 > A1 > A3 > A2. When considering a calcination rate of 60%, the average thickness of the solid solution layer followed the sequence A1 > A2 > A4 > A3, with its highest recorded value reaching 23.48 μm.

As lime with varying calcination rates interacted with different slags at 1400 °C, the average thickness of the (Ca, Mg, Fe) olivine layer formed at the lime–slag reaction interface exhibited fluctuations with the rise in calcination rate, as depicted in Figure 12. In the instances of slags A1, A2, and A4, the average thickness of the (Ca, Mg, Fe) olivine layer initially decreased, then experienced an increase, and eventually decreased once more as the calcination rate increased. The maximum values for these slags occurred at a calcination rate of 60%, measuring 92.23 μm, 75.53 μm, and 98.18 μm, respectively. On the contrary, when examining slag A3, the average thickness of the (Ca, Mg, Fe) olivine layer displayed an initial increase, succeeded by a decrease, and subsequently another increase as the limestone calcination rate increased. The peak and valley values appeared at calcination rates of 44% and 60%, respectively, measuring 106.62 μm and 38.51 μm.

Figure 13 illustrates the changes in the average thickness of the C2S layer at the lime–slag reaction interface during the dissolution of lime with varying calcination rates at 1400 °C for a duration of 120 s. In slags A1 and A2, the average thickness of the C2S layer followed a similar pattern, showing a decrease initially, followed by an increase, and then a decrease again. At a calcination rate of 0%, the C2S layer thickness in slag A2 reached its maximum value of 11.41 μm, which was the highest among all the slags. However, at a calcination rate of 0%, only slag A3 did not generate a significant C2S layer with thickness. As the calcination rate increased, the average thickness of the C2S layer gradually increased, reaching a maximum value of 7.71 μm at a calcination rate of 84%, which was the highest among all the slags. In slag A4, the thickness of the C2S layer initially decreased with increasing calcination rate and then gradually increased. Overall, the C2S layer in slag A4 was significantly thinner compared to other slags, with a minimum value of 1.23 μm at a calcination rate of 60%.

## 4. Discussion

### 4.1. Analysis of the Dissolution Mechanism of Quicklime in Slag

The schematic representation in Figure 14 elucidates the dissolution process of quicklime in a CaO-FeO-SiO_2_-MgO-based slag system at 1350 °C. Upon introducing rapidly calcined quicklime into the slag, the liquid slag envelops the lime particles. It infiltrates the interior of the lime through cracks and micropores formed during calcination, initiating the lime–slag interface reaction [9,10,11]. Due to the swift diffusion rate of FeO within the slag, it reacts with CaO to give rise to a CaO-FeO solid solution. Once a continuous and specific thickness of the solid solution layer is established, the slower diffusion of SiO_2_ is impeded by this solid solution layer. Within the solid solution layer, CaO diffuses into the slag and engages in reactions with MgO, FeO, and SiO_2_ present in the slag. This results in the formation of (Ca, Mg, Fe) olivine and (Ca, Fe) olivine. Since magnesium olivine and iron olivine can create infinite solid solutions, the chemical formula of olivine can be simplified as 3(Ca, Mg, Fe)O∙SiO_2_. As the lime–slag interface reaction progresses, concentration gradients materialize. The concentration of FeO increases from the slag to the (Ca, Mg, Fe) olivine phase, then to the CaO-FeO solid solution, ultimately becoming concentrated in the CaO-FeO solid solution layer. This concentration disparity acts as a driving force, prompting the diffusion of FeO from the slag and the CaO-FeO solid solution layer into the (Ca, Mg, Fe) olivine phase and lime, respectively. Furthermore, the CaO concentration gradient, arising from the reaction between lime and slag, facilitates the diffusion of CaO from lime to the slag, thereby perpetuating the reaction. When the slag has a high SiO_2_ content, all the free SiO_2_ in the slag cannot react completely with the CaO provided by lime. In this case, CaO reacts with the formed (Ca, Mg, Fe) olivine, resulting in the formation of low-melting-point calcium–magnesium silicate phases containing MgO in the slag [12,13,14]. As the CaO content in the slag elevates, there is a concurrent increase in the formation of solid solutions. However, due to the relatively slow diffusion rate of SiO_2_, the quantity of (Ca, Mg, Fe) olivine within the slag gradually increases, leading to a progressive thickening of the solid solution layer. Simultaneously, lime continues to dissolve in the slag, which causes an uptick in the CaO content within the slag. This, in turn, prompts reactions with FeO, resulting in the formation of additional solid solutions and consequently leading to a decrease in the FeO content. It is important to note that CaO not only reacts with SiO_2_ to generate various silicates but also, when the FeO content is below 20%, continues to replace MgO in the calcium–magnesium silicate phase, forming a dense C2S with a melting point as high as 2130 °C [15,16]. The formation of C2S consumes the calcium–magnesium silicate phase, thereby reducing the thickness of the CaO-FeO solid solution layer. In due course, a dense layer of C2S forms on the surface of the lime particles, effectively acting as a barrier that prevents further reaction and impedes dissolution. Consequently, the quantity of undissolved lime particles within the slag continues to rise, resulting in an increase in the undissolved f-CaO content within the slag [17,18,19,20].

### 4.2. Analysis of Lime Dissolution Kinetics

The liquid-phase mass transfer coefficient (*k*) can be determined using the Formula (5) developed by Matsushima et al. [4], as follows:(5)$$-\frac{\mathrm{dr}}{\mathrm{dt}}≒k\frac{\rho_{\mathrm{slag}}}{\rho_{\mathrm{CaO}}}\times \frac{∆\left(\mathrm{mol}\%\mathrm{CaO}\right)}{100}$$
where *k* is the liquid-phase mass transfer coefficient in mm/s, $\rho_{slag}$ is the density of the slag (approximately 3.3 g/cm^3^), $\rho_{CaO}$ is the density of the lime particles (approximately 2.54 g/cm^3^), and ∆(*mol*%*CaO*) is the driving force for dissolution. It quantifies the disparity between the initial *CaO* concentration and the saturation *CaO* concentration. The ∆(*mol*%*CaO*) values for various slags are provided in Table 4. The saturation *CaO* value can be ascertained by analyzing the liquidus line within the CaO-FeO-SiO_2_-MgO phase diagram [21,22].

The mass transfer coefficients for the dissolution of lime particles with different particle sizes and different burnout rates in slags A1–A4 can be calculated by using Equation (8). The results are shown in Table 4 and Table 5.

Based on the data in Table 4 and Table 5, the average mass transfer coefficient variations of different particle sizes and calcination rates of lime in the A1 to A4 slags during 120 s of dissolution at 1400 °C can be obtained. This is illustrated in Figure 15. In accordance with Figure 15, it is evident that the mass transfer coefficient (k) exhibits a consistent increase as the slag composition shifts from A1 to A4. Moreover, the particle size of lime influences a greater mass transfer coefficient during its dissolution in the slag compared to the calcination rate of lime. The maximum average mass transfer coefficient is observed when the lime particles have a diameter of 25 mm, reaching 30.41 × 10^−6^ m/s during 120 s of dissolution in the slag. Similarly, the lime with an 84% calcination rate also exhibits the highest average mass transfer coefficient of 20.15 × 10^−6^ m/s during the same dissolution time in the slag.

## 5. Conclusions

(1)In the course of the reaction between quicklime and slag, several key transformations take place. These include the development of a CaO-FeO solid solution, the formation of (Ca, Mg, Fe) olivine, and the generation of low-melting-point calcium–magnesium silicate minerals containing MgO. As the FeO content in the system diminishes, CaO engages in reactions with the (Ca, Mg, Fe) olivine, leading to the creation of a high-melting-point and compact C2S layer. This C2S layer effectively acts as a barrier, preventing the further dissolution of lime.(2)The dissolution rate of lime with different particle sizes in slags A1 to A4 generally shows an increasing trend at 1400 °C, reaching the maximum dissolution rate in slag A4. However, the dissolution rate of lime with different calcination rates in slags A1 to A4 initially increases, then decreases, and finally increases again. Therefore, it is necessary to select lime with an appropriate particle size and combustion rate to complete a rapid dissolution before A4 to avoid residual CaO entering the slag.(3)At 1400 °C, lime with a particle size of 20 mm exhibits the maximum average thickness of the C2S layer during 120 s of dissolution in slag A2, which is 12.11 μm. Lime with a calcination rate of 0% shows the maximum thickness of the generated C2S layer, which is 11.41 μm, during 120 s of dissolution in slag A2.(4)The average mass transfer coefficient for lime with different particle sizes and calcination rates during dissolution in slags A1 to A4 at 1400 °C gradually increases. The maximum values are achieved with a particle size of 25 mm and a calcination rate of 84%, measuring 30.41 × 10^−6^ m/s and 20.15 × 10^−6^ m/s, respectively.

## Figures and Tables

**Figure 1 materials-16-06487-f001:**
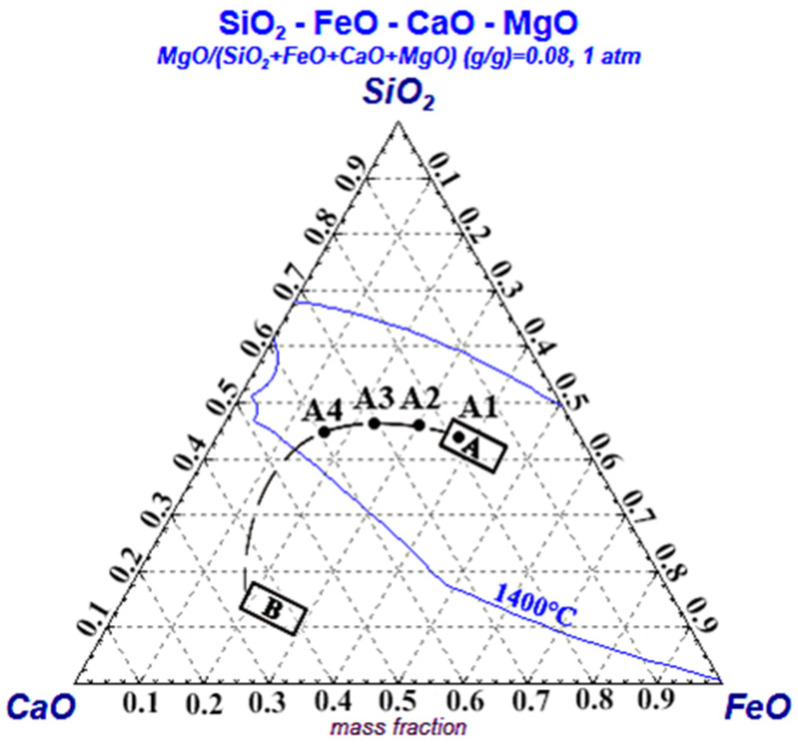
Spatial locations of slags A1–A4 in the phase diagram at 1400 °C.

**Figure 2 materials-16-06487-f002:**
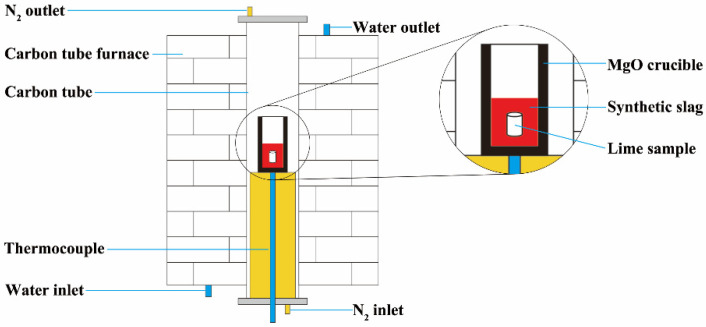
Schematic diagram of experimental setup.

**Figure 3 materials-16-06487-f003:**
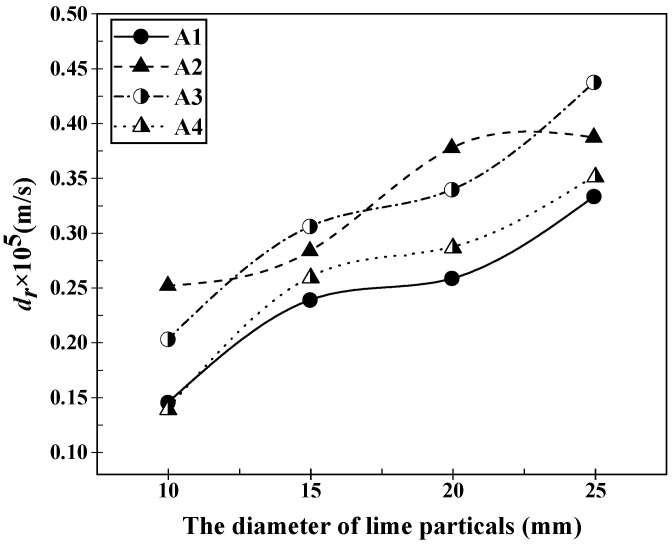
Dissolution rates of lime particles with different diameters in slags A1–A4 after 120 s of dissolution at 1400 °C.

**Figure 4 materials-16-06487-f004:**
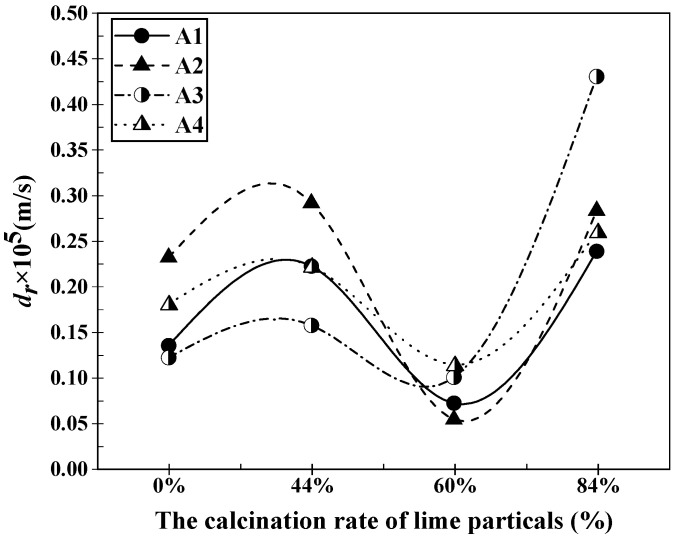
Dissolution rates of lime particles with different calcination rates in slags A1–A4 after 120 s of dissolution at 1400 °C.

**Figure 5 materials-16-06487-f005:**
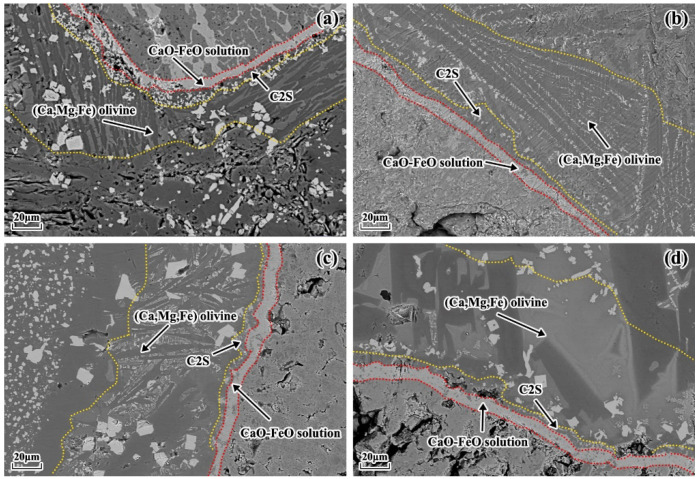
Microstructure of the lime–slag reaction interface at 1400 °C after 120 s of reaction for different particle sizes of lime and different slags. (**a**) Slag A1 with 10 mm lime particle size. (**b**) Slag A2 with 25 mm lime particle size. (**c**) Slag A3 with 25 mm lime particle size. (**d**) Slag A4 with 20 mm lime particle size.

**Figure 6 materials-16-06487-f006:**
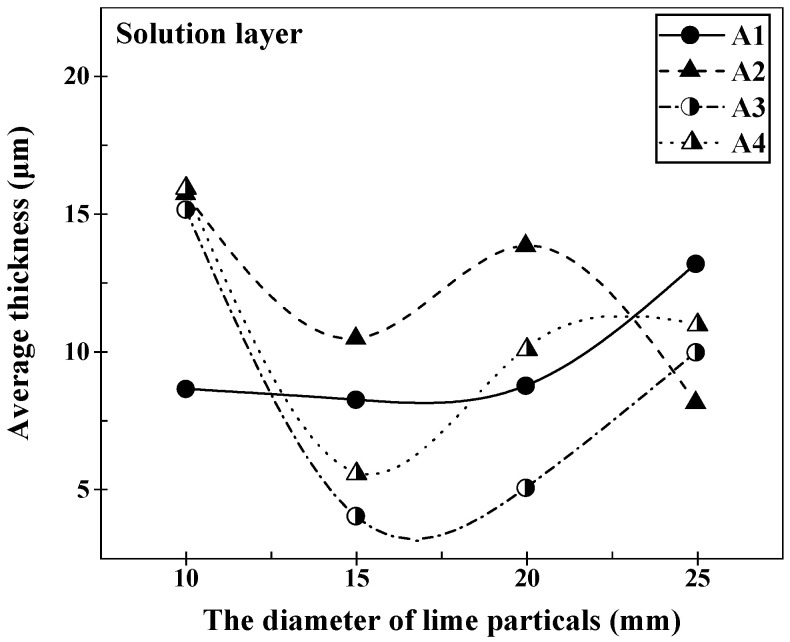
Variation in the average thickness of the solution layer at the lime–slag reaction interface during the dissolution of lime with lime particles of different diameters at 1400 °C for 120 s.

**Figure 7 materials-16-06487-f007:**
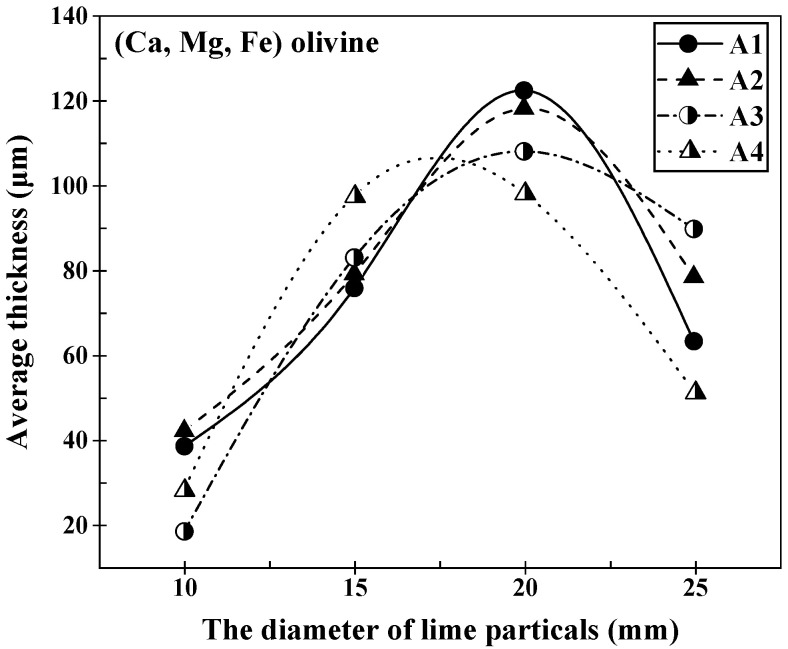
Variation in the average thickness of the (Ca, Mg, Fe) olivine layer at the lime–slag reaction interface during the dissolution of lime with different diameter of lime particles at 1400 °C for 120 s.

**Figure 8 materials-16-06487-f008:**
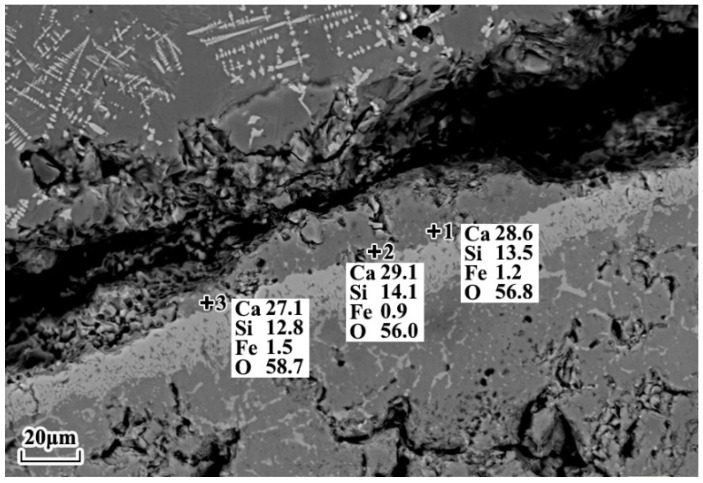
SEM-EDS point mapping results of the lime–slag interface during the reaction between 10 mm lime particles and slag A3 at 1400 °C for 120 s.

**Figure 9 materials-16-06487-f009:**
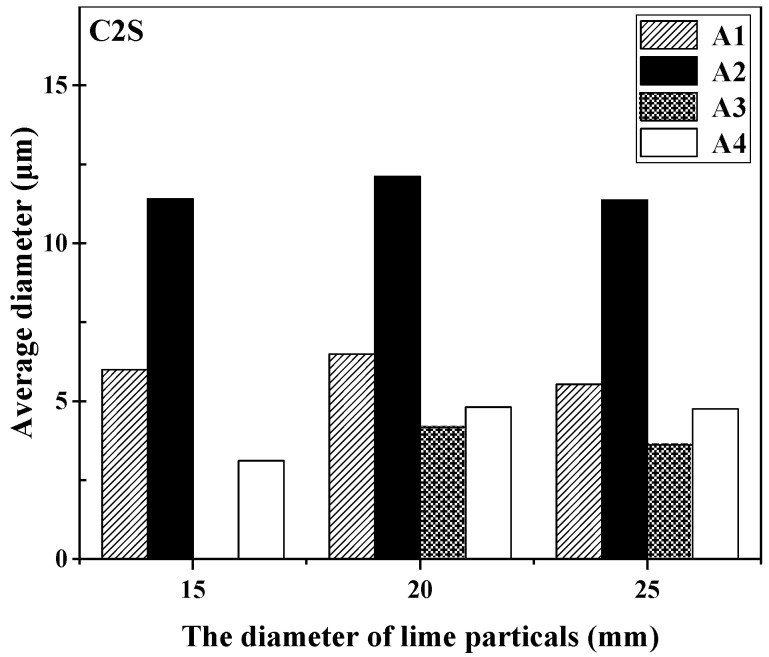
Variation in the average thickness of the C2S layer at the lime–slag reaction interface during the dissolution of lime with different diameter of lime particles at 1400 °C for 120 s.

**Figure 10 materials-16-06487-f010:**
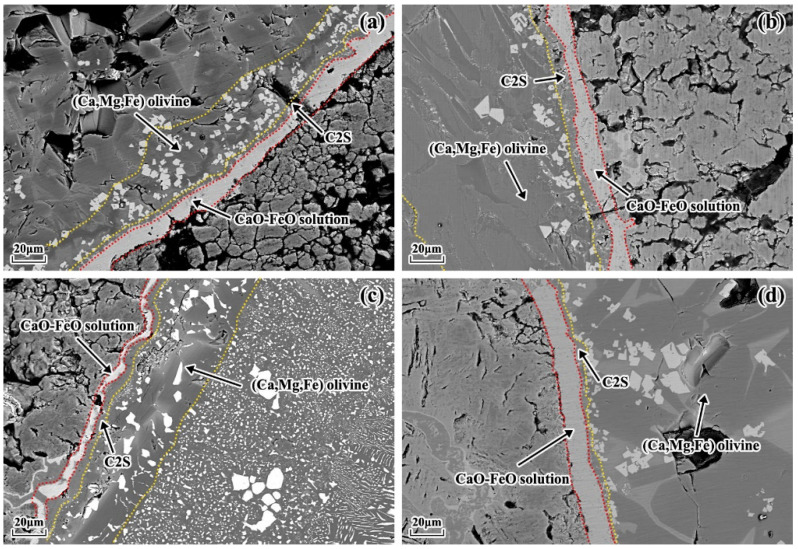
Microstructure of the lime–slag reaction interface at 1400 °C after 120 s of reaction for different calcination rates of lime and different slags. (**a**) Slag A1 with 44% calcination rate of lime. (**b**) Slag A2 with 60% calcination rate of lime. (**c**) Slag A3 with 84% calcination rate of lime. (**d**) Slag A4 with 44% calcination rate of lime.

**Figure 11 materials-16-06487-f011:**
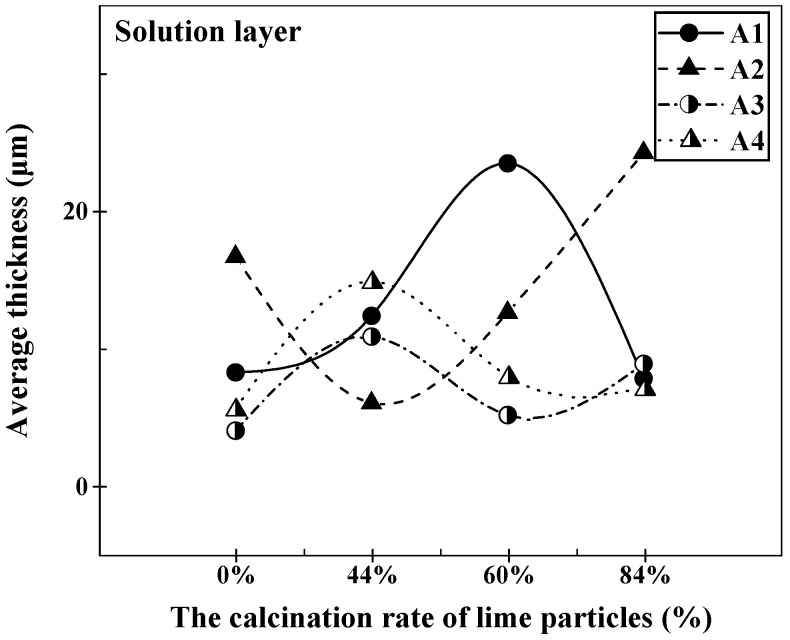
Variation in the average thickness of the solution layer at the lime–slag reaction interface during the dissolution of lime with different calcination rates at 1400 °C for 120 s.

**Figure 12 materials-16-06487-f012:**
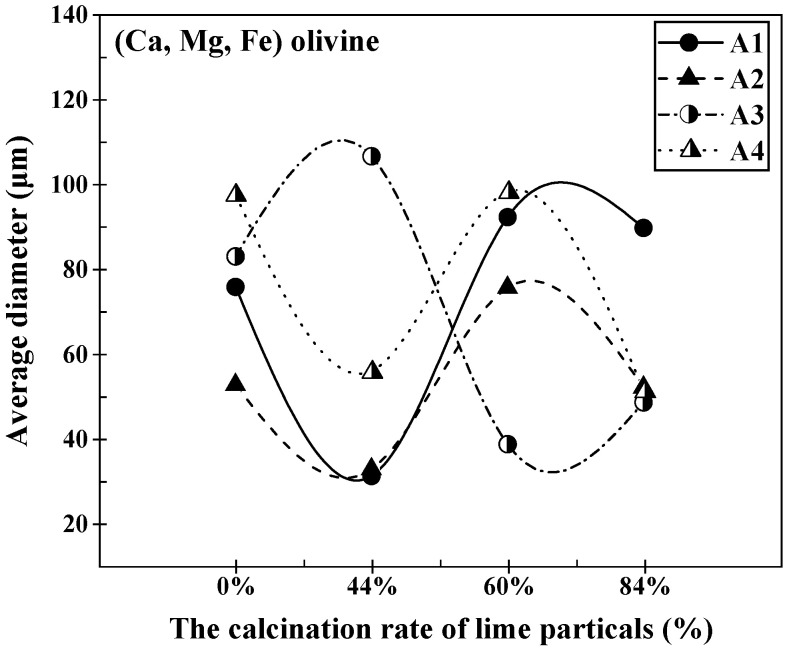
Variation in the average thickness of the (Ca, Mg, Fe) olivine at the lime–slag reaction interface during the dissolution of lime with different calcination rates at 1400 °C for 120 s.

**Figure 13 materials-16-06487-f013:**
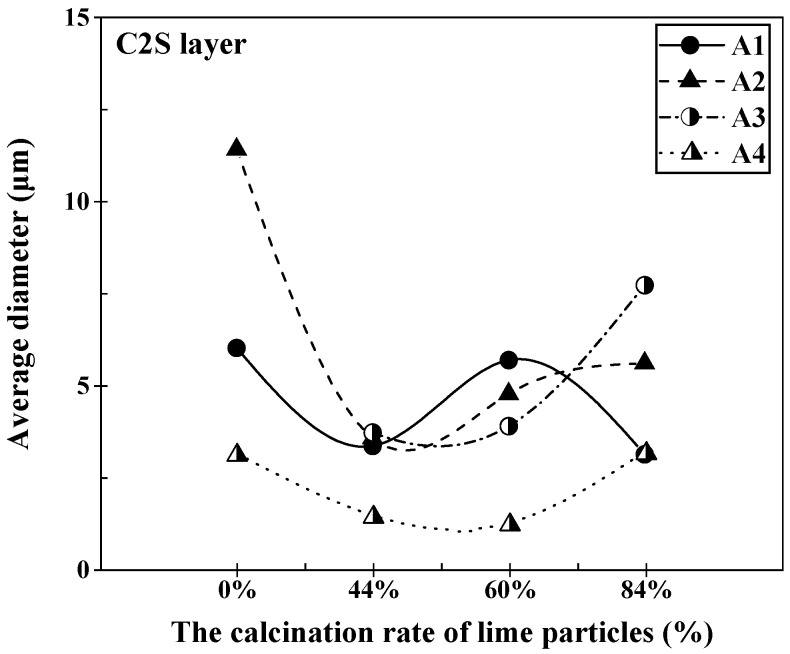
Variation in the average thickness of the C2S layer at the lime–slag reaction interface during the dissolution of lime with different calcination rates at 1400 °C for 120 s.

**Figure 14 materials-16-06487-f014:**
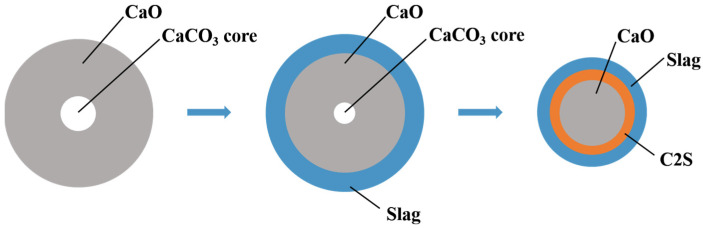
Schematic diagram of the dissolution process of lime in slag.

**Figure 15 materials-16-06487-f015:**
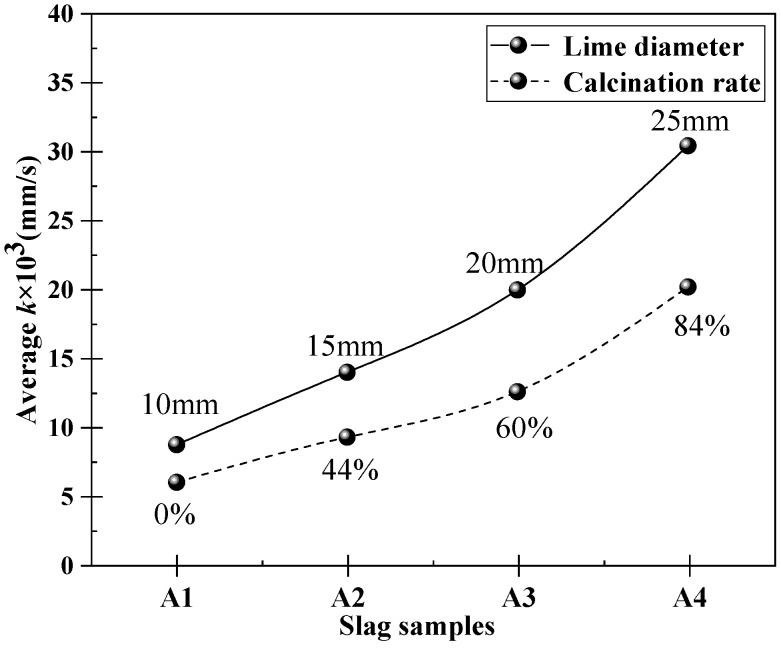
Change of average mass transfer coefficient when lime with different particle sizes and calcination rates dissolves in slag A1 to A4 at 1400 °C for 120 s.

**Table 1 materials-16-06487-t001:** Chemical composition ratios (%wt) of the slags.

	CaO	SiO_2_	Fe_t_O	MgO
A1	17	40	35	8
A2	20	44	28	8
A3	26	44	22	8
A4	37	39	16	8

**Table 2 materials-16-06487-t002:** Radius reduction in lime with different particle sizes dissolved in slag for 120 s (r × 10^5^ m).

	Slag	A1	A2	A3	A4
Size					
10 mm		3.46	6.03	4.85	3.34
15 mm		5.71	6.79	10.33	6.22
20 mm		6.19	9.06	8.14	6.88
25 mm		7.99	9.29	13.49	8.43

**Table 3 materials-16-06487-t003:** The radius reduction in lime dissolved in slag with different calcination rates for 120 s (r × 10^5^ m).

	Slag	A1	A2	A3	A4
Rate					
0%		5.71	6.79	10.33	6.22
44%		1.70	1.28	2.39	2.73
60%		5.32	6.99	1.76	5.31
84%		-	5.55	2.90	4.32

**Table 4 materials-16-06487-t004:** Mass transfer coefficients of different particle sizes of lime dissolved in slag A1 to A4 for 120 s (k × 10^6^ m/s).

	Slag	A1	A2	A3	A4
Size					
10 mm		5.15	10.80	12.55	18.85
15 mm		8.50	12.15	16.22	35.14
20 mm		9.20	16.22	21.05	38.86
25 mm		11.89	16.63	27.15	47.64

**Table 5 materials-16-06487-t005:** Mass transfer coefficients of different calcination rates of lime dissolved in slag A1 to A4 for 120 s (k × 10^6^ m/s).

	Slag	A1	A2	A3	A4
Rate					
0%		4.80	9.94	7.51	24.43
44%		7.92	12.51	9.73	30.03
60%		2.54	2.29	6.18	15.41
84%		8.50	12.15	26.72	35.15

## Data Availability

Not applicable.

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
