# Peer review of "Dissolution Behavior of Lime with Different Properties into Converter Slag"

_materials, 2023, doi:10.3390/ma16196487_

Round 1
Reviewer 1 Report
The authors provide in tadle 1 data on the chemical composition. Howeve, they do not indicate whether composition is given by synthesis or analysis. If by analysis. then it is necessary to indicate by what method the chemical analysis of substances was carried out. Classical "wet chemistry" method or using an external standard. The analyzed is mixed with other substances that interfere with is determination. The analyzed must therefore be separated from other substances before analysis. The separation process can be very difficult. and in many cases even impossible. All these points should be reflected in the comments to Table 1. On the whole, the article makes a positive impression. After comments have been made on Table 1, it may be published.
Author Response
Thanks to the reviewer for your recognition of our article, and thank you very much for carefully reading our article and putting forward valuable suggestions. We are very sorry that we failed make us description clearly, and the reviewer proposed a good question. The composition in Table 1 is the composition ratio of synthetic slag calculated by us according to the phase diagram and slag-forming route. The reagents used are all chemically pure reagents, which are calcined at high temperatures to form synthetic slag. The specific composition ratio of synthetic slag has not been chemically analyzed, for fear that readers may misunderstand this part, so this part is not explained. The composition of limestone is 98.55% CaCO3 detected by XRF. SEM-EDS is used to analyze the composition of the sample after the experiment, and its chemical composition is calculated by EDS dot. This part of the detection has been explained in the article.

Reviewer 2 Report
The paper should be rejected as it does not match the "Materials" journal level.
1. Introduction section is to big. And it's beggining should be considerably shortened. The whole chapter should be more focused.
2. The whole text about diffrent dependencies (fig. 3 - fig. 13) have no value. It just describes the figure cruves, that could be seen from the figure. But there is no explanation and the experimental approvement of it.
3. The discussion section, with it's suggesxtions about solid solution formation does not proved by any experimental investigation.
Also as an example several minor problems:
1. C3P abbreviation is not described.
2. Sentence "The limestone particles with different particle sizes are first heated for 10 minutes 126 to form lime particles" should be rephrased. Particles are obtained from particles.
3. Eq. (2) and (3) is obvious from school math course and there is no need in them.
Author Response
1), Introduction section is too big. And it's beginning should be considerably shortened. The whole chapter should be more focused.
Answer: Thank the reviewer for proposing this question. According to the review’s suggestion, in order to make the whole chapter more attractive to readers, we have rewritten to the beginning of the introduction.
2), The whole text about different dependencies (fig. 3 - fig. 13) have no value. It just describes the figure cruves, that could be seen from the figure. But there is no explanation and the experimental approvement of it.
Answer: We are very sorry that we made the reviewer have such doubts, and we are very grateful to the reviewer for pointing out my mistake. For Figures 3 to 13, we have experimental verification. However, due to the large number of experiments, the amount of data in one point corresponds to a set of experiments. If you include experimentally verified pictures in the article, it will greatly lengthen the length of the article. Therefore, we have selected representative SEM images and put them in the paper, as shown in Fig. 5 and Fig. 10.
3), The discussion section, with it's suggestions about solid solution formation does not proved by any experimental investigation
Answer: We are very sorry that we failed make us description clearly, and the reviewer proposed a good question. In my previous articles, I have elaborated on the mechanism of forming solid solutions. If the reviewers are interested, they can read my article, “Characterization of interfacial structure between quicklime and CaO-SiO2-FetOslag”.
4), C3P abbreviation is not described.
Answer: C3P has been rewritten as the full name in the article.
5), Sentence "The limestone particles with different particle sizes are first heated for 10 minutes 126 to form lime particles" should be rephrased. Particles are obtained from particles.
Answer: Thanks to the reviewer for proposing an effective suggestion. According to your opinion, and in order to avoid this kind of confusion for the readers, we have rewritten this sentence.
6), Eq. (2) and (3) is obvious from school math course and there is no need in them.
Answer: Thank the reviewer for proposing this question. Although Eq. (2) and (3) is obvious, in order to fully understand the latter equations, we think it is necessary to include these two formulas in the article.

Reviewer 3 Report
In this research, authors have attempted to investigate the evolution of the lime-slag interface and changes in the dissolution rate of lime using prepared slag and quicklime under different slag conditions during a 120-second dissolution period in the path of CaO-based slag formation. I believe that the authors did their best to achieve the above-mentioned purpose and did it in excellent level. So, congratulations. I read it several times and in my opinion, the authors should be encouraged for this article. I will briefly mention some of the advantages of this article.
The authors explained all the details in all the laboratory steps. In addition, they interpreted the results well, which shows their high knowledge in this field. In other words, they are experts. Good details are written on the microscope images. The discussion section is also well explained. Eventually, it is accepted to publish in Materials as the present form.
Author Response
Thank you very much for the review experts' recognition of our article, we will continue to work hard in the follow-up research.

Reviewer 4 Report
Manuscript entitled "Dissolution Behavior of lime with Different Properties into Converter Slag" is very interesting scientific work. The Article is well prepared with sufficient description of methods and materials followed with well described presentations of the results.
There are some minor issues that should be revised before publishing:
- Abstract could be shorter.
- references should be revised according to the template.
- The Conclusion section contains only the main findings and analyses of the results. However it should contain the summery of the research contains the most important "result/conclusion" (not analyses and the perspectives or ways for further investigation. (now this part is another paragraph of analyses of the results. )
Author Response
1), Abstract could be shorter.
Answer: According to your comment, the abstract has been revised.
2), References should be revised according to the template.
Answer: According to your reminder, the references have been revised according to template.
3), The Conclusion section contains only the main findings and analyses of the results. However, it should contain the summery of the research contains the most important "result/conclusion" (not analyses and the perspectives or ways for further investigation. (now this part is another paragraph of analyses of the results.
Answer: Thanks to the reviewer for your effective proposal after reading the article carefully.
According to your suggestion, we rewrite the results. In the future, we will pay more attention to avoid such mistakes. The revised results are as follows:
(1) During the reaction between quicklime and slag, the formation of CaO-FeO solid solution, (Ca, Mg, Fe) olivine, and low-melting-point calcium-magnesium silicate minerals containing MgO occurs. As the FeO content decreases, CaO reacts with the (Ca, Mg, Fe) olivine to form a high-melting-point and dense C2S layer, which will prevent the lime from dissolving further.
(2) The dissolution rate of lime with different particle sizes in slags A1 to A4 generally shows an increasing trend at 1400°C, reaching the maximum dissolution rate in slag A4. However, the dissolution rate of lime with different calcination rates in slags A1 to A4 initially increases, then decreases, and finally increases again. Therefore, it is necessary to select lime with appropriate particle size and combustion rate to complete rapid dissolution before A4 to avoid residual CaO entering the slag.
(3) At 1400°C, lime with a particle size of 20mm exhibits the maximum average thickness of the C2S layer during 120 seconds of dissolution in slag A2, which is 12.11μm. Lime with a calcination rate of 0% shows the maximum thickness of the generated C2S layer, which is 11.41μm, during 120 seconds of dissolution in slag A2.
(4) The average mass transfer coefficient for lime with different particle sizes and calcination rates during dissolution in slags A1 to A4 at 1400°C gradually increases. The maximum values are achieved with a particle size of 25mm and a calcination rate of 84%, measuring 30.41×10-6m/s and 20.15×10-6m/s, respectively.

Round 2
Reviewer 2 Report
Unfortunatly only a minor changes appeared. The text still describes the trends, which is seen from the figures.
The sentencse with "Particles are obtained from particles with diffrent particle size" is still should be repharsed.
No major revision for the better data representation was made.
Author Response
Dear Editor and Reviewers:
Sincerest thanks for your useful comments and suggestions on our manuscript which help us improve it to be a better scientific level. We have modified the manuscript accordingly.
Reviewers' comments:
Reviewer #2:
1), Unfortunately only a minor changes appeared. The text still describes the trends, which is seen from the figures.
The sentence with "Particles are obtained from particles with different particle size" is still should be rephrased.
No major revision for the better data representation was made.
Answer: Thanks to the reviewer for your recognition of our article, and thank you very much for carefully reading our article and putting forward valuable suggestions. We are very sorry that we failed make us description clearly, and the reviewer proposed a good question. The sentence with “Particles are obtained from particles with different particle size” has been revised as “Initially, all lime particles, regardless of their size, were subjected to a 10-minute heating period to facilitate initial formation.”